# Dangers of Mixed Martial Arts in the Development of Chronic Traumatic Encephalopathy

**DOI:** 10.3390/ijerph16020254

**Published:** 2019-01-17

**Authors:** Lucas J.H. Lim, Roger C.M. Ho, Cyrus S.H. Ho

**Affiliations:** 1Department of Psychological Medicine, Yong Loo Lin School of Medicine, National University of Singapore, Singapore 119074, Singapore; lucas.lim@mohh.com.sg (L.J.H.L.); pcmrhcm@nus.edu.sg (R.C.M.H.); 2Biomedical Global Institute of Healthcare Research & Technology (BIGHEART), Yong Loo Lin School of Medicine, National University of Singapore, Singapore 117599, Singapore; 3Centre of Excellence in Behavioral Medicine, Nguyen Tat Thanh University (NTTU), Ho Chi Minh City 70000, Vietnam; 4Faculty of Education, Huaibei Normal University, 100 Dongshan Road, Huaibei 235000, China

**Keywords:** mixed martial arts, chronic traumatic encephalopathy, neuropsychiatric sequelae

## Abstract

Chronic traumatic encephalopathy (CTE) was first discovered in professional boxers after they exhibited memory impairments, mood and behavioral changes after years of boxing. However, there is now a growing acceptance that CTE can develop in athletes of other sports due to the repetitive head trauma they receive. We present a case of a middle-aged male who presented with worsening memory, poor concentration, and behavioral changes for a year. On further cognitive testing, it was revealed that he had difficulties with short-term memory and processing speed as well as difficulties in organizing and multitasking. He had been practicing mixed martial arts (MMA) for 10 years, and later was an instructor of the sport. Through a detailed examination of his history, it was discovered that he sustained recurrent minor head concussions due to his line of work. To date, there has been limited large-scale research on head trauma in MMA. There is thus an urgent need for more studies in this area as CTE can be a chronic and debilitating illness with incapacitating neuropsychiatric sequelae. This case highlights the importance of public awareness of the risks of MMA and the dangers it poses to the brain, especially with more young people being attracted to this sport.

## 1. Introduction

Chronic traumatic encephalopathy (CTE) was first described by Martland in 1928 as “punch drunk syndrome” [1], where he hypothesized that the cognitive and behavioral symptoms observed in boxing competitors were a result of sub-lethal repeated blows to the head that the fighters sustained in their careers. It was subsequently termed “dementia pugilistica” by Millspaugh in 1937 [2]. Millspaugh noticed that the disease was characterized by memory disturbances, executive dysfunction, mood and behavioral changes, and neurological abnormalities after repetitive brain injury. Corsellis et al. [3] found that dementia pugilistica is a neuropathologically distinct condition from other neurodegenerative conditions after he presented a case series of 15 ex-boxers, which included abnormalities of the septum pellucidum associated with fenestration and forniceal atrophy, cerebellar and scarring of the brain, substantial nigral degeneration, and the occurrence of neurofibrillary tangles in the cerebral cortex and temporal horn areas.

The clinical syndrome of CTE is a combination of symptoms caused by lesions affecting the pyramidal, extrapyramidal, and cerebellar systems [4]. The cognitive and behavioral symptoms associated with CTE are reflective of the regions that have been pathologically determined to be most affected by CTE. Early cognitive symptoms primarily include learning and memory impairments. Mood changes include symptoms of depression, apathy, irritability, and suicidal thinking [5]. Behavioral issues include poor impulse control and increased aggression. Dementia would ensue in all older cases with advanced stage CTE, 10 to 20 years after retirement from the ring [6,7]. Other motor clinical signs include dysarthria in 90% of cases associated with gait ataxia. Many patients also complained of persistent headaches. Sometimes they might exhibit fine tremors, but extrapyramidal signs are rare [8]. Neuropsychological testing for former boxers suspected of having CTE has revealed difficulties in memory, information processing and speed, finger tapping speed, attention and concentration, sequencing abilities, and frontal executive functions such as planning, organization, reasoning, and judgement [6,7,9]. It has also been shown that all individuals with neuropathologically confirmed CTE cases have had repetitive head trauma [10]. 

Although most cases of CTE are found in people practicing martial arts, CTE has also been found in others with a history of repeated concussive injuries from sports ranging from professional hockey players [11], American football players, and military men in active line of duty, to the case of a circus clown who was repeatedly shot out of a cannon [12,13,14]. American football players with no diagnosis or history of concussions, but who played in positions subjected to the greatest exposure of repetitive trauma to the head, have also been neuropathologically confirmed to have CTE. This suggests that repetitive sub-concussive trauma might also lead to the development of this kind of neurodegenerative disease [15]. High-profile ex-National Football League (NFL) players such as Aaron Hernandez, Junior Seau, Dave Duerson, and Andre Waters had post-mortem confirmation of CTE after rulings that their deaths were the result of suicide [16]. These suicides could possibly be related to mood and behavioral changes resulting from CTE.

In recent years, mixed martial arts (MMA) has been heavily scrutinized by a number of medical associations which have reservations about the safety of the sport due to participants receiving repeated head trauma, with some calls for the sport to be banned completely. This is a growing concern in MMA as fighters become more aware of the risks, though there has been a paucity of mainstream reporting. In 2012, Gary Goodridge was the first MMA professional fighter to be diagnosed with CTE. 

MMA is a full-contact, no holds barred mixed combat sport, combining unarmed Oriental styles of martial arts (e.g., judo, karate, muay thai, jiu-jitsu) with Western combat techniques (e.g., Greco-Roman wrestling, boxing, kickboxing) [17]. Two contestants, wearing minimal protective equipment, skillfully adopt a combination of striking and grappling techniques, both on the ground and standing, against their opponent. A contestant attains victory by concussing an opponent into a defenseless position through blunt head trauma (knockout (KO)), disabling an opponent through joint subluxation, dislocation or soft tissue trauma, causing syncope by way of a neck choke, or coercing an opponent into submission by any permutation of the preceding [18]. A technical knockout (TKO) occurs when the contestant is unable to safely defend himself, leaving himself totally defenseless. Compared to other kinds of martial arts, MMA is a relatively new sport which started in the 1980s, and MMA competitions were first introduced in the United States (US) with the Ultimate Fighting Championship (UFC) in 1993. It was dubbed as brutal, with no holds barred, no time limits, no weight classes, and few rules. MMA was banned for a period of time in the US during the mid-1990s after it faced heavy criticism from politicians concerned about the safety of the sport, with then-US Senator John McCain calling it “human cockfighting”. The “Unified Rules” was introduced thereafter, which stipulates a list of prohibited acts (such as groin attacks, biting, throat attacks, head butting) to ensure the safety of participants [19]. These rules were subsequently adopted by most international MMA competitions. With the increased media attention of MMA worldwide, MMA is gaining popularity amongst the masses. MMA gyms are sprouting at an accelerated pace, with more amateurs taking on the sport. There is an urgent need for participants of this sport to be aware of the potential health hazards MMA poses to them. Additionally, there is currently no definite pharmacological treatment for CTE, though there have been animal studies suggesting the beneficial effects of anti-dementia drugs such as memantine [11].

Here, we illustrate a case of possible CTE in an MMA fighter with its chronic neuropsychiatric sequalae.

Written informed consent for the publication was obtained from the patient for this case report.

## 2. Case Presentation

In May 2010, a 40-year-old Caucasian man with adulthood-onset epilepsy came to our clinic for worsening memory and poor concentration for 1 year. He had progressive cognitive impairment, specifically short-term memory loss, word-finding difficulties, slower processing speed, and difficulties in organizing and multitasking. There was no reported change in his mood with no signs of depression or anxiety. He was a university graduate without family history of dementia or past history of addiction. Being an avid MMA fan, he had been practicing the sport for over 10 years. He was previously in the US Marines before working as an MMA school manager and instructor for 5 years. Recurrent minor head concussions and transient asphyxiation episodes were common in his course of martial arts training and work. On physical examination, he had hand tremors with fine motor incoordination and lower limb ataxia. Laboratory investigations, lumbar puncture, and electroencephalography revealed normal results. Magnetic resonance imaging of the brain, however, showed mild asymmetry in the parahippocampus structures with the left hippocampus appearing slightly smaller and dilatation of the left temporal horn. A neuropsychological assessment conducted in 2010 showed above-average performances on most cognitive domains except timed working memory tasks (see Table 1).

Since September 2010, he had worked as an English teacher, teaching his native language. Two years later, he could no longer stay in the job due to worsening memory and planning difficulties. He was also noted to be more irritable, with increased fatigability and distractibility. He was given methylphenidate (60 mg per day) to improve his attention. Furthermore, he developed benzodiazepine dependence but managed to undergo detoxification successfully. Repeated neuropsychological assessment in 2013 revealed worsening performance across most cognitive domains with significant decline in auditory and visual attention and memory, and further deterioration in executive function (see Table 1). The clinical and neuropsychological findings suggested chronic traumatic encephalopathy (CTE). Memantine was subsequently added to his treatment schedule and he continued to be followed up in clinic. His cognitive state deteriorated progressively and he was eventually lost to follow-up.

## 3. Discussion

A systemic review and meta-analysis of the epidemiology of injuries in MMA revealed that head injuries accounted for the highest distribution of injuries by anatomic region, with data ranging from 67.5% to 79.4% [19]. The authors also found that the injury pattern in MMA was quite similar to that of professional boxing, unlike other combat sports such as judo [20] or taekwondo [21], where blows to the head are outlawed. It is concerning that head injuries account for the highest proportion of injuries sustained by the competitor during the bout, and this became even more worrying after video analysis of 844 telecasted UFC MMA bouts revealed that about 90% of TKOs were a result of repetitive strikes. When the TKOs secondary to repetitive strikes were examined further, the 30 seconds before match stoppage was characterized by the losing competitor being hit by a series of multiple strikes to the head that increased in frequency [22]. Few would argue that when a contestant experiences a KO, he would meet the criteria for concussion, which is a type of traumatic brain injury. 

Our patient exhibited symptoms of CTE secondary to repetitive sub-concussive brain trauma received from both training and competitions. Furthermore, chronic intermittent hypoxia brought about by choking and grappling of the neck, which are commonly used in MMA, may contribute to neuronal dysfunction via reduction in cerebral tissue oxygenation. Thus, despite increased safety measures and no major head injuries sustained, MMA can still cause serious and permanent disability. In our patient, we used cognitive assays as a surrogate measure to track how severe his condition was. However, the assays were done only at baseline and 3 years later. Care should be taken in analyzing these results as there might be other contributory factors to the deterioration of the results, such as his mood state on the day of the tests, poor effort, or merely having an “off day”.

It has always been postulated that assault to a young brain which is still developing and more vulnerable to injury may have more catastrophic consequences later in life [23]. There is growing evidence that there are changes occurring in the brains of young athletes at a cellular level when exposed to mild closed head impact injury. A recent study in 2018 found that astrocytosis, myelinated axonopathy, microvascular injury, perivascular inflammation, and phosphorylated tau protein pathology in the post-mortem examination of four teenage athletes who sustained close impact sports-related head injuries and died in the acute-subacute period between 1 day and 4 months. The study also included four teenage controls who died from other medical conditions, and none of the post-mortem brains from the control group demonstrated any microscopic evidence suggestive of CTE pathology [24].

To further substantiate their point that closed head impact injury induces early CTE brain pathologies, they performed experiments on un-anaesthetized mice. They created an instrument which would allow a restrained mouse to experience a non-lethal single lateral blow to the head in the temporal-zygomatic region, and compared them to controls. They observed that the mice suffered minimal or no acute neurological impairment after the blow. The authors then performed post-mortem examinations on the brains of mice which were subjected to the lateral head impact at different time periods post-impact. By 2 weeks post-injury, there were regions of decreased neuronal density, resolving astrocytes and microgliosis, and clusters of perivascular hemosiderin-laden macrophages. They also found perivascular accumulation of phosphorylated tau proteins in the brains of mice that received the blows, which is pathognomonic for CTE [24]. It is highly likely that our patient also had these microscopic brain changes at the time of presentation. Although the study only subjected the mice to a single blow, it is evident that there were CTE changes to the brain. People practicing MMA could be subjected to repetitive blows to their head over the course of their career and hence it is possible that they would have accumulated a higher degree of phosphorylated tau proteins in the brain. In addition, when competitors continue to strike their opponents, causing them to lose consciousness and fall to the ground, they could sustain secondary head trauma when their head hits the ground [22]. With the growing trend of young people taking up MMA, it is possible that an increase in the incidence of people having CTE will be seen in the future. 

Mice models were also conducted to compare the effects of different injury time intervals on the brain [25]. Weights were dropped onto the heads of mice in two separate intervals: five hits in 3 days (short protocol) with an intervening few hours of rest before each hit, versus five hits in 15 days (long protocol) with an intervening 3 rest days between the hits. Mortality rate was significant in the short protocol at 37.5%, while it was 0% in the long protocol. It was also found that there was a higher degree of upregulation of neuroinflammatory markers following the short protocol, namely prolonged astrocytic response and the immediate activation of microglia. The upregulation of astrocytes and microglia among other inflammatory markers may be the most important cause of post-traumatic neurodegeneration as it is associated with greater phosphorylated tau pathology found in CTE [26]. On a practical note, this provides evidence that MMA athletes should have longer mandatory rest periods for recovery, especially if they have received concussive blows to the head.

It is interesting to note that there is literature [27] comparing how head injuries are sustained between American professional footballers and professional boxers. A boxer may sustain repetitive head impacts with relatively high translational and rotational acceleration incurred over a long period of time, which may cause the tearing of structures such as the septum pellucidum and substantia nigra, as well as damage to the cerebellum and cerebral hemispheres, all of which are seen in patients with CTE. In the case of professional football players, due to the difference in the biomechanics of how they receive blows on their head, they are more likely to receive cerebral concussions, which result in their being removed from play or having limited in-game activity for a short period of time. Footballers do not sustain frequent repetitive blows to the brain on a regular basis compared to boxers. It was also found in trials involving dummies that a punch produces larger rotational and translational acceleration than a tackle in football due to its larger effective radius, thereby increasing the chance of chronic brain damage [27]. Although there are no studies comparing the biomechanics of head injury in MMA participants, the pattern of injuries of boxers and MMA fighters are similar. Thus, one may postulate that MMA fighters may receive similar punches as do boxers, should they sustain head injuries, thereby elevating the chance of chronic brain damage.

What is perhaps little discussed is the role of asphyxia in the contribution towards long-term behavior and memory changes in the MMA athlete over time. As mentioned, a neck choke is identified as the cause of match stoppage when a competitor submits or the referee stops the match, as the afflicted competitor appears to be syncopal or asphyxiating [18]. One of the more common neck choke methods is called the “rear naked neck choke” which has the potential to end a match quickly. It is essentially an attack to your opponent’s back, using your legs to wrap around the opponent’s waist, and using your elbow to hook around the neck of your opponent while applying pressure from your forearm and biceps to their neck, thereby causing momentary asphyxiation or causing the opponent to pass out from the choke. In the course of an MMA athlete’s career, it is certain that they would receive such transient asphyxiation episodes multiple times from participation in matches, or even during training, given the fact that the neck choke is a commonly accepted move of offence. Neurological injury due to compression of the neck could potentially occur. In studies pertaining to suicidal hanging, a force of 2 kg was found to be sufficient to compress the jugular veins to the point of causing cerebral edema, followed by the carotid arteries with 5 kg of force, which might cause hypoxic brain injury. Compression of the airways needs a greater force of about 15 kg, which leads to severe hypoxia and death [28]. Doppler sonography reveals that it is possible to completely stop the blood flow of the carotid and vertebral arteries in a neck choke hold, which is characterized by pressure on lateral parts of the neck [29]. The issue of hypoxic ischemic brain injury (HI-BI) may develop in the long term in MMA athletes as they are subjected to frequent repeated transient asphyxiation and strangulation, leading to intermittent hypoxic events to the brain. Common mechanisms involved in the development of HI-BI include cardiopulmonary arrest, respiratory failure, and carbon monoxide poisoning. It is also known that about 30%–60% of patients who develop HI-BI as a result of cardiac arrest will develop persistent cognitive, behavioral, and neurological problems [30]. Impairment in attention, particularly vigilance and processing speed, together with memory problems have been observed in survivors with HI-BI. In addition, there are also reports of visual spatial dysfunction, apraxia, agnosia, and affective and personality changes in patients who had HI-BI [30]. In our patient, we performed repeated neuropsychological testing, which revealed decreased performance of his attention span and memory over time. There is a possibility that the patient could also have suffered some degree of HI-BI in addition to CTE, which reduced his overall cognitive abilities.

Pharmacological treatment for the neurobehavioral sequelae of traumatic brain injury appears to be limited, let alone for CTE. Despite the significant number of studies on the drug treatment of neurobehavioral sequelae after traumatic brain injury, the available evidence does not support any treatment and there are limited guidelines due to methodological problems [31]. There has been some anecdotal evidence to suggest that a non-competitive N-methyl-D-aspartate (NMDA) receptor antagonist such as memantine helps in neuroprotection after traumatic brain injury. Researchers were able to demonstrate that there are some benefits to using memantine for repetitive mild traumatic brain injury using mice models at the histopathological level and improving functional outcomes [11]. However, there are no conclusive randomized controlled trials on human subjects at present. 

There are several learning points to this case, in addition to the point that a comprehensive history that includes any form of head injuries should be investigated for young patients presenting with new neurological symptoms and comorbidities such as behavioral changes and substance abuse. The prevention against CTE is of utmost importance since there is currently no definite treatment once CTE develops, and the benefits of medications such as acetylcholine esterase inhibitors, NMDA receptor antagonists, and stimulants or neuro-rehabilitation are inconclusive. It is therefore important to discuss how we can prevent the development of CTE in practitioners of MMA. Hutchinson et al. [22] made several good recommendations to potentially improve the safety of the sport. This includes stopping the match momentarily every time a competitor is knocked down to allow time to identify and evaluate effects of traumatic brain injury. Moreover, referees should be trained to spot defenseless competitors or those who have lost consciousness and to stop the fight immediately. Lastly, after a TKO or KO, there should be medical staff on the ground to assess the athlete involved, and medical suspensions should be imposed by the relevant athletic commission to ensure adequate time for recovery prior to the next bout.

With the rising popularity of contact sports and MMA, the risk of CTE cannot be underestimated. Our current understanding of the risk of acquiring CTE is primarily limited to a biased sample of patients who were most exposed to repeated head injuries, such as professional boxers, football players, and war veterans. Little is known about the risk of CTE to individuals who are exposed to head injuries on a less frequent basis. There are also no conclusive studies to show the amount of exposure to head injury before the development of CTE. As CTE research tends to be sensationalized in the news and media, and misinterpreted by the lay public, caution needs to be exercised when discussing the results of scientific studies and generalizing these results to the population as a whole [32]. What is clear is that repetitive head injuries are definitely a contributory factor to the development of CTE, and hence extreme caution must be emphasized when participating in MMA activities. Health authorities and physicians should be aware of the links between MMA and CTE, so that cases of CTE can be better detected for the patient to receive prompt treatment. There should be greater awareness made by MMA organizations to alert the public and practitioners of the sport to the dangers the sport poses. As MMA as a whole is still a relatively young sport compared to other contact sports such as boxing or American football, there is a paucity of data on the mechanisms of head trauma in MMA. More research is certainly needed in this area, which will assist in the formulation of safeguards for the sport.

## Figures and Tables

**Table 1 ijerph-16-00254-t001:** Comparison of neuropsychological assessment over 3 years.

Test	2010			2013			Change
	Raw Score	Scaled Score/Norm Mean (SD)	Classification	Raw Score	Scaled Score/Norm Mean (SD)	Classification	
Auditory Attention							
Letter Number Sequencing *	14	14	Above Average	9	8	Average	↓
Visuo-Spatial Attention—Spatial Span *							
Forward	9	11	Average	4	4	Borderline	↓
Backward	10	14	Above Average	5	7	Below Average	↓
Total	19	13	Above Average	9	4	Borderline	↓
Auditory Memory—Logical Memory *							
Immediate Recall	59	15	Superior	35	9	Average	↓
Learning Slope	7	13	Above Average	5	10	Average	↓
Delayed Recall	40	16	Superior	22	9	Average	↓
Recognition	-	-	-	26	10	Average	
Auditory Memory—Word List *							
Immediate Recall	41	14	Above Average	27	6	Below Average	↓
Learning Slope	5	10	Average	4	8	Average	=
Delayed Recall	12	16	Superior	8	12	Average	↓
Recognition	-	-	-	21	7	Below Average	
Visual Perceptual/Construction Abilities							
Block Design **	36	9	Average	48	12	Average	=
Rey–Osterrieth Complex Figure Copy	35	32.31 (2.67)	Average	31	32.31 (2.67)	Average	=
Rey–Osterrieth Complex Figure Recall	24	16.56 (6.69)	Above Average	19.5	16.56 (6.69)	Average	↓
Processing Speed							
Symbol Digit Modality	41	52.96 (9.26)	Below Average	39	52.96 (9.26)	Below Average	=
Trail Making Test A	41 ***	-	-	27	28.54 (10.09)	Average	
Trail Making Test B	48	58.46 (16.41)	Average	49	58.46 (16.41)	Average	=

* Wechsler Memory Scale—Third Edition; ** Wechsler Adult Intelligence Scale—Third Edition; *** 1 pencil lift and 1 error as he could not find a number which he had covered with his finger.

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
