# Peer review of "Dangers of Mixed Martial Arts in the Development of Chronic Traumatic Encephalopathy"

_ijerph, 2019, doi:10.3390/ijerph16020254_

Round 1
Reviewer 1 Report
This paper is comprehensive, informative and a delight to read. There is a highly informed discussion and demonstrated knowledge of MMA and the potential neurological consequences.
Subsequently I have very few recommendations to the authors, except that perhaps they refer to the mechanistic effects of repetitive mild traumatic brain injury more.
In particular the growing animal literature showing that forces of impact that show no obvious signs or symptoms with one impact may generate pathology if repeated over time.
Additionally the discussion between timing of impacts is important. For example in MMA and boxing bouts are infrequent (often not for many weeks between bouts) but have severe impacts, whilst most team-based sports have lower impact forces but are more frequent (i.e. weekly matches).There is a growing and substantive animal model literature on how time between impacts, and frequency of mild traumatic brain injury affects neurological and neurodegenerative outcomes.
Author Response
Dear Reviewer
thank you for your kind comments
1)Perhaps they refer to the mechanistic effects of repetitive mild traumatic brain injury more. In particular the growing animal literature showing that forces of impact that show no obvious signs or symptoms with one impact may generate pathology if repeated over time.
Reply: yup i agree with this point. There are actually no obvious signs and symptoms in animal models. Most of the studies were post mortem brain analysis.
2) Additionally the discussion between timing of impacts is important. For example in MMA and boxing bouts are infrequent (often not for many weeks between bouts) but have severe impacts, whilst most team-based sports have lower impact forces but are more frequent (i.e. weekly matches).There is a growing and substantive animal model literature on how time between impacts, and frequency of mild traumatic brain injury affects neurological and neurodegenerative outcomes.
Reply: sure, i will include a short discussion on this.
Reviewer 2 Report
This is a well written manuscript. However, the introduction needs a few more references for certain claims such as pathology reports from non Football players cases. Also, The Discussion the point about NMDA agonists could be brought up and cited in the introduction.
The cognitive assays were reliable. However, with only 2 repetitions, we have no way of knowing if the effects were due to the impacts or if this one person just had an “off day” (as we know people can vary day to day in cognitive abilities). This caveat should mentioned in the discussion.
Author Response
Dear reviewer, thank you for your kind comments.
1)The introduction needs a few more references for certain claims such as pathology reports from non Football players cases.
Reply: Sure, i will include reference from a systemic review from all pathological cases.
2) Also, The Discussion the point about NMDA agonists could be brought up and cited in the introduction.
Reply: sure, i can put in a small point on that in the start
3) The cognitive assays were reliable. However, with only 2 repetitions, we have no way of knowing if the effects were due to the impacts or if this one person just had an “off day” (as we know people can vary day to day in cognitive abilities). This caveat should mentioned in the discussion.
Reply: i agree with your point as well.